# Beyond the Unitary State: Multi-Level Governance, Politics, and Cross-Cultural Perspectives on Animal Welfare

**DOI:** 10.3390/ani14010079

**Published:** 2023-12-25

**Authors:** Paul Chaney, Ian Rees Jones, Nivedita Narayan

**Affiliations:** Wales Institute of Social and Economic Research (WISERD), SPARK, Cardiff University, Maindy Road, Cardiff CF24 4HQ, Wales, UK; jonesir4@cardiff.ac.uk (I.R.J.);

**Keywords:** animal welfare, law, multi-level governance, civil society, devolution, political cultures, policy divergence

## Abstract

**Simple Summary:**

Existing cross-cultural research on animal welfare often overlooks the way that policy and law are not the exclusive domain of central government. This can result in an over-simplification or misrepresentation of the true situation. The political culture and institutional arrangements for governing the modern state are more complex than a “one-size-fits-all” approach. It is argued that cross-cultural research needs to give greater attention to differences *within* as well as between unitary states. Specifically, it needs to examine developments in constituent nations and territories. Here we illustrate this by drawing on new research in the United Kingdom, and examine how ‘devolution’—or decentralized government for Wales and Scotland—is providing contrasting opportunities for NGOs, campaigners, and the public to lobby to improve animal welfare policy based on local practices, beliefs, and demands (collectively known as the “political culture”). Our findings show how this is important because it results in contrasting animal welfare policies and laws in the constituent nations of the UK.

**Abstract:**

It is argued that extant cross-cultural research on animal welfare often overlooks or gives insufficient attention to new governance theory, civil society, politics, and the realities of devolved or (quasi-)federal, multi-level governance in the modern state. This paper synthesizes relevant social theory and draws on new empirical findings of civil society accounts of campaigning on animal welfare policies and law in the United Kingdom. It is presented as a corrective to arguably reductive, earlier unitary state-based analyses. Our core, evidence-based argument is that cognizance of civil society activism and the contrasting institutional governance structures and political cultures of constituent nations in unitary states—such as the UK—are providing opportunities for the territorialization of legally grounded animal welfare regimes, and culturally distinctive practices.

## 1. Introduction

Existing cross-cultural research on animal welfare often focuses on unitary states and overlooks the way that animal welfare policy and law are not the exclusive domain of central government. The political culture and institutional arrangements for governing the modern state are often more complex than a “one-size-fits-all” approach. Accordingly, it is argued that cross-cultural research needs to give greater attention to differences *within* as well as between unitary states. Specifically, it needs to examine developments in constituent nations and territories. Thus, the literature gap this study aims to fill is the lack of attention to governance within a state due to an assumption that states are culturally homogenous. Failure to do this can result in an over-simplification or misrepresentation of the true situation. Here we illustrate this by drawing on new research in the United Kingdom, and examine how ‘devolution’—or decentralized government—for Wales and Scotland is providing contrasting opportunities for “civil society” (viz. charities and NGOs, campaigners, and the public) to lobby for improved animal welfare based on local practices, beliefs, and demands. The UK is a state, but animal welfare is largely a devolved issue to the four constituent nations; thus, Wales, Scotland, England, and Northern Ireland are effectively unitary states for animal welfare, not the UK. Our findings show how the post-devolution governance shift is important because it results in contrasting animal welfare policies and laws in the constituent nations of the UK. This is based on our interviews with animal welfare activists and NGOs and secondary data analysis of their campaigning ‘grey literature’ [1] and parliamentary proceedings.

The following discussion addresses two principal research questions based on developments in Wales and Scotland: (1) What are the views and experiences of civil society campaigners lobbying for animal welfare gains? And (2) What evidence is there of the “territorialization” of animal welfare policy driven by civil society activism? As we shall see, the findings show how *inter*-state analysis is important to cross-cultural research on animal rights because it furthers understanding of the different political opportunity structures for advancing animal welfare in the constituent nations of the UK and advances our knowledge of how the lobbying and activism of local policy communities results in contrasting animal welfare policies and laws in the devolved nations.

To address the foregoing questions, the remainder of this paper is structured as follows: first, we briefly outline the research context and explain why the UK is a suitable case study. Next, relevant strands of social theory are synthesized to make the conceptual case for cross-cultural analysis to examine differences *within* as well as between states. In the following section, the research methodology is summarized. Subsequently, the research findings are discussed. The concluding section reflects on the significance of the results.

## 2. Research Context

Extant cross-cultural research on animal welfare often overlooks the way that animal welfare policy and law are not the exclusive domain of central government [2,3,4]. This can result in an over-simplification if it fails to examine the issue from a multi-level governance perspective. The United Kingdom provides an example of this. The UK is a political creation based on the union of England with Wales (in 1536), Scotland (in 1707), and Ireland (in 1801, subsequently, Northern Ireland in 1921). It is a comparatively recent example of the ‘devolutionary trend [that] has swept the world… [involving widespread] transference of power, authority, and resources to subnational levels of government’ (Rodriguez-Pose and Gill 2003, p. 334) [5]. Yet it was not until 1998/99 that elected legislatures were established for Wales, Scotland, and Northern Ireland. Prior to that juncture, animal welfare policy and law were largely determined by Westminster for Wales and England. Scottish legislation has always had a different legal base and often passed different laws to England and Wales on animal welfare. In the Twentieth Century, minor variations to Westminster legislation on animal welfare were introduced by the territorial ministries of the UK government: the Welsh and Northern Ireland Offices [5]. After 1999, prior to Brexit, Wales and Scotland passed their own devolved legislation on animal welfare implementing EU Regulations and Directives. The UK leaving the EU has meant that Wales, Scotland, and England are able to pass laws on issues such as farm animal welfare that had previously been the exclusive preserve of the EU. It is also the case that, post-Brexit, Northern Ireland remains a member of the EU via the Single Market and Customs Union and it implements EU legislation on animal experiments, not Westminster. As earlier analysis has revealed, in the face of lobbying by Welsh and Scottish policy communities, the post-1999 devolved era has seen the rise of political competition as parties have advanced distinctive policies in their manifestos for Welsh and Scottish parliamentary elections (Chaney 2022) [6]. While limited aspects of animal welfare powers remain reserved to Westminster (e.g., the use of animals in scientific experimentation in Great Britain), the majority are devolved and as we shall see, the past quarter century has seen increasing territorialization of animal welfare policy and law at the devolved level.

## 3. The Theoretical Case for a Multi-Level Governance Approach

The aim of this part of the present discussion is to synthesize relevant strands of social theory to make the conceptual case for future cross-cultural analyses of animal welfare to examine differences *within* as well as between states. Our starting point is the concept of ‘new governance’. Rod Rhodes’ original characterization (1997: 53) of it posits that it should be conceived of as a system characterized by interdependence between organizations. Governance (or the structures and processes of government) is broader than government alone. Crucially, it involves different tiers of government and includes attention to the actions of non-state actors. This matters, for the changing form of the state in the modern era means that the boundaries between public, private, and voluntary sectors are ever shifting and often opaque. For non-state actors, there is a significant degree of autonomy from the state. Activists, social movements, and campaigners—such as those concerned with advancing animal welfare—are not accountable to the state; they are self-organizing. Although the state does not occupy a privileged, sovereign position, it can indirectly and imperfectly steer networks [7]. Crucially, Rhodes highlighted new forms of governing and state structures caused by devolution or government decentralization. This he referred to as ‘the hollowing out of the state’ (Rhodes 2007: 1258) [8]. The result he concluded was a ‘differentiated polity’ (Rhodes and Weller 2005) [9] (i.e., one not solely predicated on administration by central government but where constituent nations may have their own governance arrangements). All these factors are important to cross-cultural analyses of animal welfare (and other policy issues) because they underline the need to look beyond analysis of central government policy and explore regional structures of governance and the role of civil society. As Rhodes (2007: 1258) proceeds to explain, this perspective ‘opens new avenues of exploration on key issues confronting policymaking and policy implementation… including: the sectoral character of policymaking; the mix of governing structures; the “philosopher’s stone” of central coordination; devolution to the constituent territories of the UK… [in short, we can] use it to develop a new way of seeing state authority in its relationship to civil society’. It is a point more recently articulated by Bob Jessop (2015: 20), who argues that ‘state powers… are activated through changing sets of politicians and state officials located in specific parts of the state apparatus, in specific conjunctures… the state’s structural powers and capacities cannot be fully grasped by focusing on the [central] state alone’ [10]. Accordingly, ‘multi-level governance’ (or alternatively, ‘multi-spatial governance’) aligns with the concept of new governance and emphasizes the tiered—or multiple layers—governance that exists in nominally unitary states [11]. As Rhodes’ (2017: 23) also underlines, ‘governance poses questions about the shifting boundaries between state and civil society’ [12].

Put simply, civil society is the space outside of the family, business, and the state (Cohen and Arato 1992) [13]. However, it should be noted that ‘these elements are intertwined such that their boundaries are effectively seamless’ (Eto 2012: 78) [14]. Thus, civil society comprises a diverse range of associational activities extending beyond the family, to encompass non-governmental organizations (NGOs—or alternatively, civil society organizations—or CSOs), pressure groups, charities, community groups, social movements, and campaigning organizations [15]. A broad literature attests to the contested nature of ‘civil society’ (Edwards 2004) [16]. For example, De Tocqueville (1835) distinguishes between ‘political’ society and civil society, asserting that civilian and political associations act as a counterbalance to liberal individualism and the state [17]. Later Twentieth Century thinkers such as Gramsci (1948) [18] and Habermas (1962) [19] also underlined the fluidity of relations between the state and civil society. Notably, they claimed that the relationship is not unidirectional, but that civil society both resists *and reinforces* hegemonic ideas—whether about economic and social life—or, as in the present case, human-non-human relations. Subsequently, writers such as Putnam (1993) [20] stressed how the production and accumulation of social capital is integral to the effective functioning of society and democracy. Social capital here refers to the ‘features of social organization, such as trust, norms, and networks, that can improve the efficiency of society by facilitating coordinated actions’ (Putnam 1993: 167) [20]. As Hindess (2004: 40) explains, governance and governmentality ‘should be understood more widely than the supreme authority in states, [but instead] as action aimed at influencing the way individuals regulate their own behaviour… the aim of modern government of the state is to conduct the affairs of the population in the interests of the whole. This is not restricted to the government but is performed also by agencies in civil society’ [21]. As noted, the key point in all of this is that state decentralization or ‘devolution’ (in the present case, to Wales and Scotland) redefines governance and (re-)creates polities (or territorial political systems) for constituent nations and regions in unitary states. These are headed by their own governments and parliaments and provide political engagement opportunities for civil society activism to promote animal welfare claims and campaign for legal and policy change at the devolved level based on local needs and demands. International examples include (but are not limited to) regional indigenous legislatures in the Russian Federation [22], as well as in European Union members states, such as the Autonomous Communities (Regions) of Spain, Regions of Italy, German Federal States such as Bavaria, and the Autonomous Regions of the Azores [23], as well as the Canadian provinces and territories (notably Quebec and First Nations) [24].

Allied to the foregoing, classic social movements theory underlines the importance of what are termed “political opportunity structures” [25]. These are the structures and mechanisms that allow civil society organizations to press their policy claims on government [26]. They are allied to the academic literature on neo-institutionalism [27]. In short, this shows us how the design and operation of political institutions shapes the policy process. As will be seen, our findings suggest that devolution has made governance more “structurally allowing” for the territorial animal welfare lobbies in Wales and Scotland.

Lastly, governance and civil society are integral to the literature on “political culture”. The term was initially proposed by Gabriel Almond [28] and subsequently developed with Sidney Verba [29]. As Stephen Chilton explains, it refers to the reciprocal and iterative relationship between institutions and governance structures and culture in a society by noting ‘the term promised to solve in a scientific, cross-culturally valid way the micro-macro problem: the classic problem of specifying how people affect their political system, and vice-versa’ [30]. Underlining the present argument that current research needs to adopt a multi-level governance approach to cross-cultural studies of animal welfare, a burgeoning literature traces how devolution in the UK: (1) was born out of the historical and cultural differences between the Welsh, Scottish, English, and Northern Irish peoples [31], (2) how Wales and Scotland possess their own policy communities on animal welfare [32], and (3) how in its wake, devolved politics are shaping distinctive laws, policies, and cultural practices in each nation [6].

## 4. Methodology

This study adopts a mixed methods approach. It is based on the findings of 30 semi-structured interviews with civil society organizations (or ‘NGOs’—non-governmental organizations) that campaign for animal welfare (25 were undertaken in the post-Brexit period 2022–2023 and 5 were completed in 2013, the latter offering a longitudinal perspective). These were selected using purposive sampling to reflect the diversity of the sector and span different animal welfare fields including: wildlife, agriculture, and companion animals. Interviews were conducted in Wales, Scotland, and England [33]. Northern Ireland was not included because of the extended periods over recent years when devolved governance was suspended owing to political impasse (and direct rule from Westminster was imposed). The research interviews were transcribed and analyzed thematically. This dataset was complemented by secondary data analysis of the ‘grey’ literature of NGOs [34] (including newsletters, social media and websites), as well as public domain records of government policy, law, and parliamentary proceedings.

## 5. Results

### 5.1. What Are the Views and Experiences of Civil Society Animal Welfare Campaigners? How Does ‘Devolved’ Lobbying Compare to Campaigning at Westminster?

A key finding from our series of in-depth interviews with campaigners representing civil society organizations (CSOs) is their frustration with Westminster and generally positive views of the opportunities to engage and lobby parliamentarians in Wales and Scotland. Speaking about Westminster, this interviewee reflected the consensus: “The government definitely just can’t prioritize anything above [and beyond] Brexit anymore. And so, I think that’s probably the biggest problem we’ve got is that Brexit absorbs the oxygen from anything that isn’t Brexit”. Allied to this, civil society campaigners expressed their fury and frustration at the Conservative government’s decision on 8 June 2023 to withdraw the Animal Welfare (Kept Animals) Bill [35]. This was going to deliver pledges set out in the Conservative Party’s 2019 manifesto for greater protection of animals kept, imported into, and exported from the UK, including ending the unlicensed captivity of primates and export of live animals for slaughter and fattening. The rapid turnover of UK prime ministers and the political volatility of Westminster are also seen by CSOs as a key problem. As this interviewee explained: “Scotland’s bit more [politically] stable [than Westminster]. And that’s why we’ve had more success in getting [policy gains] … [for example] they did put in new regulation for [dog] rescues… because they are working to a pretty regular schedule of elections, and it’s a little bit more stable. It’s a devolved administration… [that] feels better and more efficient”.

In a similar vein, this interviewee spoke of the difference between Cardiff and Edinburgh on the one hand, and Westminster on the other: “I think because some animal welfare is devolved. It can be easier to deal with—[i.e.,] you know the devolved governments. I think when it comes to dealing with the UK Government it’s always harder. They are quite good at stonewalling or shutting things down”. This was a view shared by several interviewees, including this manager: “I think the devolved governments and devolved parliaments are much easier to engage with, much more open and are far more democratic. So, we sit on cross-party groups [of parliamentarians and external organizations] in both Wales and Scotland, and we’ve had success engaging with the Scottish Government relatively easily, so that, I mean, *they* actually approached us to give feedback and a heads up on launching a public consultation, which was quite a surprise! … I think in comparison to the UK Government, it feels much more trusting of [civil society] organizations”.

As noted, in conceptual terms, social movements theory and neo-institutionalism underline the importance of “political opportunity structures”, or mechanisms and processes that facilitate or frustrate civil society organizations in pressing their policy claims on government. As the foregoing quotes attest, interviewees in this study spoke of how devolution has made governance more structurally allowing for the territorial animal welfare lobbies in Wales and Scotland. Notably, a number alluded to the work of the Scottish Animal Welfare Commission. Established under section 36 of the Animal Health and Welfare (Scotland) Act 2006, the Commission has been established to provide independent advice to Scottish Ministers on the welfare of sentient animals, primarily on wildlife and companion animal welfare [36]. While in Wales, interviewees highlighted Animal Welfare Network for Wales (AWNW). Also established in 2006, this is an independent network of organizations that engages with key stakeholders, including the Welsh Government. As this interviewee explained: “Other advantages we’ve got is we’ve got Animal Welfare Network Wales, [through this…] we’ve got a network that is accessible to smaller charities… and that means that there is a framework there that they can get involved in if they want” [37]. Such collaboration continues to shape policy, as in the case of the Welsh Government’s Code of Practice for Animal Welfare Establishments [38].

Interviewees also spoke of the importance of the Third Sector Partnership Council. This is a unique feature of devolved governance in Wales that is designed to shape policy and lawmaking. It is based on a statutory requirement for collaboration in policy work between the Welsh government and the voluntary sector [39]. As this interviewee notes: “then, of course, there’s that direct link between the third sector and the obligation in law for the [Welsh Government] minister to meet with the third sector and animal welfare will be represented there through Animal Welfare Network Wales and Companion Animal Welfare Group Wales”.

### 5.2. What Evidence Is There of the “Territorialization” of Animal Welfare Policy Driven by Civil Society Activism?

The past quarter century of devolved government for Wales and Scotland has resulted in increasing policy divergence between the nations of the UK. As Table 1 illustrates, there is clear evidence of the territorialization of animal welfare owing to the contrasting protections set out in the Welsh and Scottish Governments’ policies and laws. Interviewees alluded to numerous examples of policy divergence, and we now turn to consider selected examples. In definitional terms, when we talk of “territorialization” of animal welfare, we mean civil society organizations (CSOs) successfully lobbying for distinctive laws and policies that convey contrasting legal protections in either Wales, Scotland, or England. Often, they place duties on named bodies, groups, and individuals, and/or make named practices unlawful and subject to criminal or civil proceedings and penalties, including fines and imprisonment. Whilst these new laws improve animal welfare in one jurisdiction, the practices they proscribe may remain lawful in other UK jurisdictions.

The first example of territorialization stems from civil society lobbying in Wales Table 2. It concerns the banning of snares [62]. This was achieved under the provisions of the Agriculture (Wales) Act passed by the Welsh Parliament (or *Senedd*) in 2023. Snares are wire or cord nooses that are placed in undergrowth with the intention of killing animals such as rabbits (although they are indiscriminate and catch all manner of creatures, including cats and dogs). Typically, the animal dies a slow and agonizing death as it is choked to death when trying to escape. Prior to the new enactment, CSOs lobbied for reform of the pre-devolution Westminster legislation covering Wales (and England). For example, a network of CSOs prepared an open letter to Members of the Senedd and asked their supporters to lobby for change [63]. It told its members: “Imagine a Land of Our Future where we restore nature, tackle climate change and secure healthy, sustainable food for future generations. For this to happen, we need to make sure the Agriculture (Wales) Bill is strong enough to deliver these changes. We can only influence the next phase effectively with your help” [64]. Subsequently, another CSO told its members: ‘As you may be aware, we are celebrating yet another momentous victory for animals: Wales just banned the use of snares. In a few short months, shooting estates will no longer be able to lay death traps for animals. We got what we asked for: a straightforward ban on snaring, with no amendments, no loopholes. The bill was passed last night, meaning badgers, foxes, cats and dogs will be protected from a long and agonizing death in these cruel traps. But in England, snaring remains legal. Please donate today to help us take on the rest of the UK and protect more animals from the cruelty of snares’ [65].

Our second example is a case of post-devolution policy transfer, where civil society organizations are using reforms in one jurisdiction to lobby for reform elsewhere [66]. Mandatory closed-circuit television (CCTV) was introduced in slaughterhouses in England in 2018. Amongst other things, the resulting video record reduces the chances of abuse and cruelty by abattoir staff. At present, this is not a requirement under Welsh law, although the Welsh Government is currently consulting on a ban. According to one animal welfare CSO, “polling has indicated that a huge 82% of the public in Wales supports the introduction of CCTV. It is clear that we are not alone in the belief that this is an important step in improving farm animal welfare” [67]. It proceeded to implore the public to “take action and help thousands of animals—getting this vital change is a huge reassurance that animal welfare standards are being delivered… We can’t do this alone. The Welsh Government needs to hear from you”. In the face of such demands, the Welsh Government has announced it will make CCTV mandatory from Spring 2024 [68].

Broadly defined, animal welfare policy extends beyond mammals and includes birds, reptiles, fish, and invertebrates, as well as habitat protection. Our third example of territorialization has again been driven by CSO activism. In June 2023, the courts found in favour of a welfare charity after it brought a judicial review case against the Scottish Government’s licensing of scallop dredging and seabed trawling [69]. Whilst the sentience of shellfish and crustacea is debated, the sentience of lobsters, octopus, and crabs has recently been confirmed by scientists [70]. Accordingly, this case has wider welfare significance because the dredging and trawling causes habitat destruction for these forms of marine life. Following the judicial review, these practices have been ruled unlawful in Scotland. In contrast, over the past decade, and in the face of civil society campaigning [71], the Welsh Government has introduced new regulations, including the introduction of vessel tracking devices in Welsh scallop fisheries [72]. However, current Welsh legislation falls short of a ban [73]. A public consultation on scallop fisheries is presently gathering evidence [74].

Badger culling provides a further stark illustration of legal and policy divergence on animal welfare. In England, government policy and law permit the killing of badgers in an attempt to control bovine Tuberculosis (bTB), an infectious respiratory disease which affects cattle. The UK government estimates over 200,000 badgers have been killed over recent years [75]. According to one NGO that is opposed to the policy, “The solution to bovine TB is simple: move to the better types of cattle testing that are available; switch to cattle vaccination; stop killing badgers; significantly reduce cattle movements… The [Westminster] government has deliberately not invested in cattle vaccine research” [76]. Wales looked set to adopt a similar policy until civil society activism forced a rethink. Initially, in 2009, the Welsh Government legislated to allow a nonselective badger cull [77]. In response, an animal welfare NGO sought a judicial review and the Court of Appeal ruled that the 2009 Order [allowing the cull in law] should be quashed [78]. In consequence, in 2011 the Welsh Government ruled out a badger cull and replaced it with a five-year vaccination program [79]. As a result, Welsh herds are tested and cattle movements are more strictly controlled. In further contrast to England (where animals other than cattle are not routinely tested for bTB), in Wales, badgers, deer, and other livestock species are tested and badgers are vaccinated against bTB. In Scotland, culling is unlawful. Badgers and their setts are protected under the Protection of Badgers Act 1992 (as amended by the Wildlife and Natural Environment (Scotland) Act 2011) [46]. Northern Ireland had no badger cull from 2012, but introduced one in 2022, then replaced it with a badger vaccination programme in 2023 following a campaign by civil society organizations. According to UK government data, the incidence of bTB in cattle herds in Wales and Scotland is lower than in England [80].

**Table 2 animals-14-00079-t002:** Selected Animal Welfare Manifesto pledges for the 2021 Welsh and Scottish Parliament Elections.

Wales	Scotland
“Develop a national model for regulation of animal welfare, introducing registration for animal welfare establishments, commercial breeders for pets or for shooting, and animal exhibits. We will improve the qualifications for animal welfare inspectors to raise their professional status. We will require CCTV in all slaughterhouses, we will ban the use of snares, and restrict the use of cages for farmed animals. We will not allow the culling of badgers to control the spread of TB in cattle” (Welsh Labour Party 2021, p. 37) [81].	“We will support farmers to produce more of our own food needs sustainably and to farm and croft with nature, including through enhanced animal welfare… Animal Welfare: We will adopt the highest possible animal welfare standards, including shifting to entirely free range, woodland or barn chicken and egg production. We will modernise and update the Animal Welfare Act from 2006 and implement the new livestock worrying legislation. We will seek to reflect so far as we can, new EU animal welfare labelling to promote food produced to higher than EU welfare standards. We will create a new Scottish veterinary service to ensure that we have enough people with the right qualifications in veterinary services, animal health and food safety to meet all our needs across the public and private sector for land and marine based animal health issues. We will ban live exports of animals for fattening and slaughter and only allow live transport of livestock to and from islands and the mainland with stringent welfare standards in place. We will legislate to close loopholes in the law protecting foxes and other wild mammals and remain committed to implementing the licensing of driven grouse shooting. We will implement the recommendations of the Deer Working Group and modernise deer management legislation” (SNP 2021, p. 56) [82].
“We will introduce a baseline support payment to offer the [farming] industry greater economic stability. This support will be used to encourage the highest standards of public health and animal health and welfare” (Plaid Cymru 2021b. p. 62) [83]. Animal Welfare: A Plaid Cymru Government will work with stakeholders to build upon the high level of animal welfare standards already in place in Wales. We will: • Improve the enforcement and delivery of licensing requirements relating to dog breeding establishments in Wales, building on the recent review of regulations by the Wales Animal Health and Welfare Framework Group. • Improve horse welfare by taking action on equine tethering. • Review pet vending, focusing especially on the regulation of animals sold online. • Issue model tenancy proposals on pets in social housing and work to reduce barriers between homeless pet owners and homeless shelters. • Support the development of statutory codes of practices for the keeping of exotic pets in Wales” (Plaid Cymru 2021b. p. 64) [83].	
“[We will] Enhance Animal Welfare and Protect Wildlife: • Establish an animal offender register in Wales • Ban the keeping of primates as pets • Review pet breeding standards and registration requirements to ensure adequate protection for animals and bring forward the ban on the third party sale of cats and dogs • Promote honest labelling to enhance consumer choice, including distinguishing between stunned and non-stunned slaughter methods and introduce CCTV in abattoirs • Establish a £20 million Wales Wildlife Fund to support conservation efforts across Wales” (Welsh Conservative Party 2021, p. 30) [84].	“Scottish Labour will carry out a full review of Scotland’s outdated animal welfare legislation, with a view to strengthening wildlife protection law and animal welfare… We will introduce a National Animal Cruelty Register to support enforcement agencies. We will reform the law on keeping domestic pets in different tenures after life events, including domestic violence. The pandemic has highlighted concerns over the illegal importation of puppies, and we need to raise public awareness and ban imports of very young puppies, and other illegal pets not on the positive list of species that are suitable to keep as pets. We support a more comprehensive approach to public education on animal welfare. We will introduce a comprehensive ban on fox hunting and snares and the use of electric shock collars. There also needs to be more effective monitoring of raptor conservation and stronger penalties. Labour supports a ban on live animal exports for fattening and slaughter. Parliament should pay full regard to animal welfare requirements when formulating and implementing policies” (Scottish Labour Party 2021, p. 172) [85].
“Work with the UK Government to ensure that it only enters into trade agreements under which imported goods meet the high environmental, food quality and animal welfare standards expected of home-produced food… Despite progress in recent years, there is much to be done to ensure the highest standards and protections for animals in Wales. We will: • Ensure that animal welfare standards are as least as good if not better than those we enjoyed as members of the European Union • Pass a Wildlife Act for Wales, creating clarity and consistency on the policy and legislation that protects wildlife. • Work with the sector to improve the welfare of farm animals, including live exports, and wider issues such as public sector food procurement and labelling. • Regulate all animal sales to protect the welfare of any pet traded, bought or sold in Wales. • Issue guidance to local authorities ensuring they do not use their powers under the Anti-Social Behaviour, Crime and Policing Act 2014 in a way that compromises animal welfare. • Work with Police and Crime Commissioners to take action against growing instances of dog thefts” (Welsh Liberal Democratic Party 2021, unpaginated) [86].	“Delivering the highest standards of animal welfare—The Scottish Conservatives stand for the highest standards in animal welfare. We are proud of our campaign to deliver Finn’s Law in the last Parliament, giving proper protection to service animals like policedogs. We will… bring forward a new Animal Welfare Bill… As part of our Animal Welfare Bill, we would ban the sale of dogs with cropped ears in Scotland. We would follow Wales in banning the use of electric shock collars. We would amend the Dangerous Dogs Act, so that dogs are not automatically put down due to their breed. We also would take forward measures to improve the welfare of farmed animals in transportation. In doing so, we will take account of Scotland’s geography and established farming models. We do not believe that the scheme as proposed in England is right for Scotland” (Scottish Conservative and Unionist Party 2021, pp. 42–43) [87].

As earlier work has noted, political parties’ manifesto pledges on animal welfare tell us about contemporary political thinking on the relationship between humans and non-humans, evolving notions of sentience, and how these are related to structures and processes of democratic governance. There are several contrasting conceptualizations of electoral politics, yet a core, shared feature is their underlining of the fact that elections are intimately related to civil society policy demands and opinions. However, whilst earlier work has examined the electoral politics of animal welfare in state-wide elections (Chaney 2014) [88], internationally, there has been a dearth of attention to how parties use the new political spaces associated with the electoral politics of ‘regional’ legislatures to address animal welfare issues. One exception is research on devolved elections in the UK 1998–2017. This concluded that: ‘the move to multi-level electoral politics provides new political spaces to advance animal welfare and how meso-ballots are increasingly attuned to the symbiosis of humans and animals. These factors are driving the territorialization of policy and leading to distinctive animal welfare regimes in the different countries of the UK’ (Chaney et al., 2022: 116) [6]. The present analysis of the 2021 Welsh and Scottish Parliament elections supports this and underlines how animal welfare policy is grounded in the formal representational structures and processes of contemporary democracies and the dynamic relationship between political actors, parties, civil society, and government. As Table 2 reveals, animal welfare has become a mainstream political issue in parties’ electoral competition. In Wales and Scotland, all main parties contesting the 2021 elections set out detailed and extensive policy proposals. There are multiple pledges to use Welsh and Scottish legislation to increase regulation, and evidence of proposed policy transfer (as parties cite policy developments elsewhere and call for them to be implemented in their polity). Intra-party divergence on animal welfare is also notable. This refers to ‘statewide’ parties (that is, parties that operate across the UK or GB) holding different policy positions on animal welfare in the different nations. This is because, in the wake of devolution, to varying degrees, the Labour Party, Liberal Democrats, and the Conservative Parties have moved to (quasi-)federal structures with a degree of autonomy for the Welsh and Scottish wings of these UK parties. Strikingly, on live animal transportation, the right-wing Conservative and Unionist Party in Scotland rejected the policies of its governing administration at Westminster by stating: “In doing so, we will take account of Scotland’s geography and established farming models. We do not believe that the scheme as proposed in England is right for Scotland” (Scottish Conservative and Unionist Party 2021, pp. 42–43) [87].

Following the 2021 elections, the parties forming government in Wales and Scotland—the Welsh Labour Party (in Partnership agreement with Plaid Cymru) and the Scottish National Party (SNP) in Scotland—are in the process of implementing their manifesto pledges. For example, the Animal Welfare Strategy for Wales 2021–2026 [89] reasserts the commitments to the introduction of mandatory registration for animals in private keeping; improving the qualifications of animal welfare inspectors; ensuring CCTV in all slaughterhouses, and restricting the use of cages for farmed animals. As these measures are implemented, the associated policy consultations in each nation provide further opportunities for the territorial policy lobbies of civil society organizations to shape the evolving animal welfare regimes. This can be a fractious process, as typified by the Welsh Government’s current public consultation on stricter regulation on the release of gamebirds [90]. This has been welcomed by wildlife and animal welfare NGOs, yet it has led opponents to claim it is the backdoor route to a shooting ban [91].

## 6. Conclusions

This study’s findings are significant in a number of key regards. First, the empirical data confirm the importance of devolution to civil society advocacy of animal welfare. They reveal increasing policy divergence and the territorialization of animal welfare regimes with distinctive laws and policies in Wales and Scotland. Moreover, compared to Westminster, civil society activists point to the devolved political spaces as being more structurally allowing and responsive to their claims. This corresponds to what social movements theory calls “political opportunity structures”. These are mechanisms that allow civil society organizations to press their policy claims on government (also described in the academic literature on neo-institutionalism, civil society, and multi-level governance).

In addition, our findings show how CSOs’ lobbying for policy transfer means that gains (in the form of new laws and policies) in one devolved jurisdiction may subsequently be used in campaigning for similar reforms elsewhere. This is evident in calls in Scotland to introduce a legal ban on electronic dog collars, as introduced in Wales. A further example is hunting with dogs; with bans in force on public land in many areas of Wales, there are calls to devolve powers over hunting with dogs to the Welsh Parliament in order to introduce a full ban, as in Scotland [92]. A further consequence of devolution is that improvements in welfare can take place in one jurisdiction whilst being resisted in others (e.g., the use of snares banned in Wales but lawful in England, and stricter regulation of the release of game birds in Wales). There is also evidence of cross-party working on animal welfare at the devolved level (e.g., ‘We will continue to work with other parties across the Scottish Parliament to strengthen protections for animals in Scotland by bringing forward a new Animal Welfare Bill…’ (Scottish Conservative and Unionist Party 2021, pp. 42–43) [87], as well as intra-party policy divergence in statewide parties (this is the case with all examples of distinctive animal welfare policies advanced by statewide parties in one nation that are not introduced in every polity).

As noted, this study’s relevance and rejoinder to cross-cultural studies of animal welfare is that policy and law are not the exclusive domain of central government. Sole emphasis on statewide governments can result in an over-simplification or misrepresentation of the true situation. The political culture and institutional arrangements for governing the modern state are complex and result in contrasting animal welfare policies and laws in constituent nations, states, and territories. Many of the advances seen in Wales and Scotland would not be possible under the pre-devolution “one-size-fits-all” mode of lawmaking from Westminster. This is not to argue that devolution is automatically a panacea for animal welfare. Arguably, one reason why it is easier to get laws on animal welfare in Wales and Scotland is that Westminster plays the role of two legislatures (England and UK), so it is more difficult to get a time slot there than it is in the Senedd or Scottish Parliament, and there are fewer Members of the *Senedd* (MS) (60) in Wales and Members of the Scottish Parliament (MSPs) (129) than in Westminster (660), so it is easier to lobby those legislatures. Moreover, only certain issues have been devolved (e.g., no defence or foreign policy), so arguably, it is easier to get discussion on issues that have been devolved (e.g., animal welfare) in those legislatures. It is also the case that all the distinctive animal welfare policies and practices outlined in this study may be subject to future reversal. This will be determined by the vagaries of elections, campaigning, and party politics over the following decades. Yet it is argued that to date, at least, the present findings suggest multi-level—or devolved—governance in the UK has generally been beneficial to animal welfare, for it presents opportunities for innovation and, through the process of policy transfer, may lead to a process of ‘levelling-up’ of welfare standards. The present findings also underline how devolved elections and associated civil society lobbying of parliamentarians are driving policy divergence as parties seek to respond to local demands and public opinion in their efforts to be elected. It is also the case that, in the UK at least, there is increasing evidence that animal welfare policy has moved from the margins to the mainstream of political issues. Finally, it is worth reemphasizing that, as in the present case, future cross-cultural animal welfare studies need to adopt a multi-level governance lens in order to fully understand the way that policy and law develop and diverge within unitary states.

## Figures and Tables

**Table 1 animals-14-00079-t001:** Examples of the territorialization of animal welfare law in Wales, Scotland, England, and Northern Ireland.

Topic	Wales	Scotland	England	Northern Ireland
Hunting with Dogs	Under the Hunting Act (2004) [40], it is illegal to hunt wild mammals with dogs in Wales (and England). There are exemptions that allow hunting for certain types of humane control. This is called exempt hunting [41].	Protection of Wild Mammals (Scotland) Act 2002 [42], and Hunting with Dogs (Scotland) Act 2023 [43]. Inter alia, a person who deliberately hunts a wild mammal with a dog commits an offence, as does an owner or occupier of land knowingly permitting another person to do so.	Under the Hunting Act (2004) [40], it is illegal to hunt wild mammals with dogs in England (and Wales). There are exemptions that allow hunting for certain types of humane control. This is called exempt hunting [41].	Northern Ireland remains the only part of the UK where hunting with dogs remains legal [44].
Badger culling	In Wales, the Welsh Government has ended the use of badger culling as a technique to control bovine Tuberculosis (bTB) [45].	Badgers and their setts are protected under the Protection of Badgers Act 1992 (as amended by the Wildlife and Natural Environment (Scotland) Act 2011) [46].	In England, government policy and law permit the killing of badgers in an attempt to control bovine Tuberculosis (bTB). See the Protection of Badgers Act 1992, as amended [47].	Plans by Northern Ireland’s Department of Agriculture, Environment and Rural Affairs (DAERA) to introduce an English-style badger cull in efforts to control bovine TB in cattle were thrown out by the High Court in a judgement on 25 October 2023 [48].
Licensing of animal activities	Animal Welfare (Licensing of Activities Involving Animals) (Wales) Regulations 2021 [49]. This law concentrates on one issue and makes it illegal for a commercial seller to sell a puppy or kitten they have not bred themselves at their own premises and they must ensure the mother is present. Henceforth, puppies and kittens can only be purchased from where they were bred or from a rescue or rehoming centre.	The Animal Welfare (Licensing of Activities Involving Animals) (Scotland) Regulations 2021 [50]. Introduces a more modern and flexible licensing regime for pet selling, dog breeding, and other matters, such as extending licensing to a wider range of activities involving animals than currently found in existing pre-legislation.	The Animal Welfare (Licensing of Activities Involving Animals) (England) Regulations 2018 [51], as amended [52]. This law covers a broader range of topics than its Welsh and Scottish counterparts. Anyone wanting to get a new puppy or kitten in England must now buy direct from a breeder or consider adopting from a rescue centre instead.	Welfare of Animals Act (Northern-Ireland) 2011 [53] currently falls short of a ban on third party sales of puppies and kittens [54].
	Animal Welfare (Licensing of Activities Involving Animals) (Wales) (Regulations) 2021. New licensing requirement in relation to animal exhibits and animal establishments [55].	Animals and Wildlife (Penalties, Protections and Powers) (Scotland) Act 2020. This improves the penalties and powers available to enforcement agencies and the courts. It also introduces improved procedures for making permanent arrangements for animals taken into possession by the authorities to protect their welfare [56].	Animal Welfare Act 2006 extends to England and Wales, and to some provisions Scotland and Northern Ireland [57].	Welfare of Animals Act (NI) 2011 sets out provisions—all who own, or are responsible for an animal, are required by law to care for it properly and take reasonable steps to ensure its welfare needs are met [58].
Shock collars	The Animal Welfare (Electronic Collars) (Wales) Regulations 2010. Section 2 “It is prohibited for a person to—(a) attach an electronic collar to a cat or a dog; (b) cause an electronic collar to be attached to a cat or a dog; or (c) be responsible for a cat or a dog to which an electronic collar is attached” [59].	The use of electric shock collars is currently legal in Scotland. In 2018, the Scottish Government published non-statutory guidance advising against the use of these devices and other aversive training methods [60].	The Animal Welfare (Electronic Collars) (England) Regulations 2023 will make the use of shock collars unlawful from 1 February 2024 [61].	There are currently no legal restrictions on the use or sale of shock collars in Northern Ireland.

## Data Availability

Data supporting reported results will be deposited with the ESRC data archive https://www.data-archive.ac.uk/ (accessed 18 December 2023).

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
