# Peer review of "Beyond the Unitary State: Multi-Level Governance, Politics, and Cross-Cultural Perspectives on Animal Welfare"

_animals, 2023, doi:10.3390/ani14010079_

Round 1

Reviewer 1 Report

Comments and Suggestions for Authors

Brief summary

This article examines how devolution, or government decentralization, impacts animal welfare policy and animal welfare organizations’ ability to work within political systems. The authors argue that examining differences within a state structure is an important yet missing dimension of cross-cultural work on animal welfare.

Comments

The article’s research questions are interesting and contribute to a fuller scope of knowledge about animal welfare policy. Key terminology was explicitly and clearly defined, which is especially important given the multi-disciplinary scope of Animals. The data supports the authors’ conclusions. I found the tables that were used to be helpful in following the authors’ arguments.

As a reader outside of the UK, I thought it would be helpful to bring footnote 35 (or at least part of it, into the main text at lines 191-93. Knowing that a bulk of interviews were mostly conducted in 2022-23 was significant in helping me understand the long-standing impact of Brexit for the stability of the UK government.

I have two questions that revolve around the contextualization of this study as a cross-cultural analysis.

1.       I felt like a bit more discussion about what cross-cultural analysis means in different contexts may be helpful. I agree with the authors that Scotland, Wales, and England represent distinct cultures, and this is clearly documented in the discussion (lines 183-186). That being said, I would like more discussion about the literature gap this study aims to fill. Is the lack of attention to governance within a state due to an assumption that states are culturally homogenous? If so I think that bears saying explicitly, and it need not take too long.

2.       I am also curious about how the authors see their conclusions as applying outside of cross-cultural analyses in countries such as Canada, when I am located. (Fair enough, Canada has distinct cultural groups, for example, between French and Anglophone Canadians, and with First Nations, Inuit, and Métis Indigenous peoples, not to mention Canada’s rich multicultural demographics). But when I think about the division of powers between the Federal and the Provincial and Territorial governments, I am not sure that qualifies as a cross-cultural analysis (except for Quebec and the federal government). I am curious as to whether a federalist system like Canada would constitute a type of devolution (I am thinking specifically of the description in lines 161-62).

Author Response

Please see uploaded file with our responses to the Reviewer.

Reviewer 2 Report

Comments and Suggestions for Authors

This is an excellently written and presented paper.

The only minor correction is one of clarification.  The phrase 'animal welfare rights' will be confusing the readership of Animals, particularly at the moment with so much published on 'animal rights'.  I can see how you have tentatively described it as a combination of the policy and legal protections afforded to animals, but these are quite distinct from 'animal rights', and the insertion of the word 'welfare' may not adequately convey the different meaning you propose.

Could you seek to rephrase 'animal welfare rights' as a term, or alternatively be explicit in the opening paragraph how this term is defined.  (if you opt for the definition approach, then you will need to remove animal welfare rights as a term from the abstract as it won't make sense in the abstract until the definition is understood).  I think the definition approach will be tricky because even a 'welfare policy and law protection' isn't a right of the receipient animal.  It would be best to remove this term altogether in order to prevent readers fixating on it and loosing the main thrust of this very well written paper.

Author Response

(The authors gave the same response as above.)

Reviewer 3 Report

Comments and Suggestions for Authors

This is a useful case study to look at the live experiment of how devolution aids improving animal welfare in the UK.  However the study is confused at how devolution works on animal welfare and this needs greater context.

It would also be best to show the differences if Table 1 shows the different dates and different policies with the laws in the four devolved states on certain issues eg

Hunting with Dogs

Badger Cull

Licensing of animal activities

Shock Collars

Animal welfare rights is used throughout this paper and from line 281 I understand the use of the word rights is meant to determine duties placed on bodies (line 281) but it gets very confusing in the paper as animal welfare and rights are very different things - animal welfare is based on a scientific approach to improving an animals' well being whereas animal rights is an ethical approach to stop doing certain things to animals. To avoid confusion I would always use the words animal welfare as that is what Government passes laws on and replace the word rights with the word duties eg line 8, 17, 19, 21, 25, 30, 35, 185, 275, 289.  If you are using "rights" as "duties" as outlined in line 281 this should solve confusion

line 37 the paper is very confused as to what devolution is - UK is a state but animal welfare is largely a devolved issue to the four state - so Wales is a unitary state for animal welfare not the UK - this needs to be explained in a few lines as to what devolution does for animal welfare in the UK

line 90 note that as Northern Ireland is a member of the EU via the Single market and Customs Union it implements EU legislation on animal experiments not Westminster. This line needs to be changed.

line 43 the Government is not decentralised on animal welfare - it is devolved

line 55 on animal welfare it is not intra State as the state is Wales or Scotland - it in inter State

line 80 Scottish legislation has always had a different legal base to England and Wales and so it is not true to say that devolution in 1999 changed this - Scotland has always passed different laws to England and Wales fro the 19th Century on animal welfare 

line 83 - this is  wrong - Wales and Scotland passed their own devolved legislation on animal welfare implementing EU Regulations and Directives after 1999

line 84 this fundamentally misunderstands devolution: leaving the EU did not devolve the powers to Scotland and Wales - the Government of Wales Act1999 and the Government of Wales Act 2006 and Wales Acts 2014 and 2017 provided more powers and devolution to Wales in the past 23 years.   This line needs to be amended or deleted. The UK leaving the EU had no material change at all on how animal welfare was regulated in Wales viz a viz England - it merely meant that both countries were able to pass laws on issues such as farm welfare that had previously been the exclusive preserve of the EU

Table 1 is very useful but why not contrast with the English legislation to show the differences

eg Circus Acts in all three jurisdictions (essentially the same but passed at different times)

Animal Welfare Act 2006 England and Wales

Glue traps England ban 2022 Wales 2023

LAIAR England 2018 covers 8 issues, LAIAR 2021 in Wales one issue, LAIAR 2021 in Scotland covers 7 issues etc

Hunting Act 2005 England and Wales 

Sentencing Act 2021 England and Wales 

line 320 Wales will now introduce mandatory CCTV in abattoirs in Spring 2024 (https://www.theyworkforyou.com/senedd/?id=2023-06-14.3.511921)

line 361 this is a good case study and N Ireland can be referred to here as N Ireland had no badger cull from 2012 then U turned to introduce one in 2022 then U turned again to replace it with badger vaccination in 2023 after a campaign by USPCA and Badger Trust NI 

line 420 the shock collar ban in England has passed the Lords but not the Commons - this is a very good example of the domino effect of devolved governments impact on others eg Wales passed in 2010 and Scotland now consulting in 2023 and England midway through in 2023; N Ireland has not passed any legislation and this is possible as it is a subsidiarity law so not in the acquis 

line 421 Hunting is another good example between the Hunting Act 2005 (E&W) and Hunting with Dogs (Scotland) Act 2023. 

The reason why it is easier to get laws on animal welfare in Wales and Scotland is 1) Westminster plays the role of two legislatures (England and UK) so it is more difficult to get a time slot there than it is in the Senedd or Scottish Parliament 2) there are fewer MS (60) and MSPs (129) than in Westminster (660) so easier to lobby those legislatures - this should be discussed in the discussion. As only certain issues have been devolved (eg no taxation, trade, foreign policy) it is easier to get discussion on issues that have been devolved (eg animal welfare, education) in those legislatures. 

Comments on the Quality of English Language

English good

Author Response

(The authors gave the same response as above.)
